# ASYMMETRIC LEARNING DYNAMICS FOR REACHING STACKELBERG EQUILIBRIUM

## ABSTRACT

The Stackelberg equilibrium, a cornerstone of hierarchical game theory, models scenarios with a committed leader and a rational follower. While central to economics and security, finding this equilibrium in dynamic, unknown environments through learning remains a significant challenge. Traditional multi-agent learning often focuses on symmetric dynamics (e.g., self-play) which typically converge to Nash equilibria, not Stackelberg. We propose a novel and provably convergent framework based on *asymmetric learning dynamics*. In our model, the leader employs a reinforcement learning (RL) algorithm suitable for non-stationary environments to learn an optimal commitment, while the follower uses a no-regret online learning algorithm to guarantee rational, best-response behavior in the limit. We provide a rigorous theoretical analysis demonstrating that this asymmetric interaction forces the time-averaged payoffs of both agents to converge to the Stackelberg equilibrium values. Our framework corrects several flawed approaches in prior analyses and is validated through a comprehensive set of experiments on canonical matrix and Markov games.

## 1 INTRODUCTION

Hierarchical decision-making is ubiquitous, appearing in domains ranging from market competition and cybersecurity to supply chain management and international relations (Von Stackelberg, 2010). The Stackelberg game model (Stackelberg, 1934) provides the foundational framework for these scenarios, designating one agent as a *leader* who commits to a strategy first, and another as a *follower* who observes the leader's commitment and plays an optimal best response. The resulting solution concept, the Stackelberg equilibrium (SE), often yields a higher utility for the leader compared to the simultaneous-move Nash equilibrium (Nash Jr, 1950).

Despite its importance, the question of how agents can *learn* to play a Stackelberg equilibrium in a general-sum Markov game (Littman, 1994) without full knowledge of the environment is far from solved. Most multi-agent reinforcement learning (MARL) research (Bowling & Veloso, 2002) has focused on symmetric learning dynamics, such as self-play, where all agents use the same algorithm. These dynamics are well-suited for finding Nash equilibria in symmetric games but are generally not guaranteed to converge to the hierarchical Stackelberg solution. Existing methods that do target SE often rely on strong assumptions, such as full differentiability of the game dynamics or the follower's ability to compute an exact best response in a single step (Fiez et al., 2020).

This paper addresses this gap by proposing and analyzing a novel **asymmetric learning dynamic (ALD)** for reaching Stackelberg equilibrium. Our central idea is to equip the leader and follower with fundamentally different, yet complementary, learning algorithms that mirror their roles in the hierarchy:

- The **Leader** employs a reinforcement learning (RL) algorithm (e.g., PPO (Schulman et al., 2017)) capable of handling non-stationary environments. This makes the leader's optimization landscape dynamic with continuously adapting follower's policy.

- The **Follower** employs a *no-regret* online learning algorithm (e.g., Hedge (Freund & Schapire, 1997)). This guarantees that, over time, the follower's average behavior is indistinguishable from that of a perfectly rational agent playing a best response.

The intuition is that the follower's no-regret guarantee provides a stable, predictable signal of rationality. The leader's RL algorithm, in turn, learns to exploit this emergent rationality to find the optimal commitment strategy. We provide rigorous theoretical guarantees for this dynamic, proving that the time-averaged payoffs of both agents converge to their respective Stackelberg equilibrium values. Our main contributions are:

1. We propose a novel asymmetric learning framework (RL-Leader, No-Regret-Follower) for finding Stackelberg equilibria in general-sum Markov games.

2. We provide a comprehensive and rigorous theoretical analysis proving that this dynamic converges to the Stackelberg equilibrium. Our proofs correct several logical flaws and imprecise arguments found in prior theoretical sketches.

3. We outline a set of experiments designed to validate our theoretical findings in canonical matrix and Markov games, comparing our approach against relevant baselines.

## 2   RELATED WORK

**Existing Approaches for Direct Learning of Stackelberg Equilibria**   Learning a Stackelberg Equilibrium (SE) in unknown environments presents a distinct challenge that falls outside the scope of traditional multi-agent reinforcement learning (MARL). Standard MARL paradigms, such as self-play, are designed for symmetric dynamics and typically converge to Nash equilibria (Gerstgrasser & Parkes, 2023), failing to address the asymmetry inherent in Stackelberg settings (Bai et al., 2021). While recent works have extended MARL to accommodate leader-follower dynamics, they often depend on predefined hierarchies or heuristic assumptions and lack formal guarantees of convergence to a Stackelberg equilibrium (Foerster et al., 2017). To address this gap, several recent studies have proposed methods to learn Stackelberg equilibria directly. One prominent line of work involves gradient-based methods, which leverage differentiable models of the game and opponent behavior to compute equilibria (Balduzzi et al., 2018; Sakaue & Nakamura, 2021). The primary limitation of these approaches, however, is their restriction to continuous or differentiable action spaces, which makes them ill-suited for the many discrete or non-differentiable environments found in practice. Other works explore learning dynamics in hierarchical or repeated games but often lack rigorous convergence proofs or rely on strong, often unrealistic, assumptions about the agents' learning protocols (Ozdaglar et al., 2021; Arslantas et al., 2025).

**No-Regret Learning as a Foundation for Stackelberg Dynamics**   No-regret learning offers a powerful and theoretically grounded framework for modeling leader-follower interactions. As a foundational tool in game theory (Cesa-Bianchi & Lugosi, 2006; Freund & Schapire, 1997), its central principle is that an agent's average regret—the difference between its accumulated loss and that of the best single action in hindsight—vanishes over time. Classic algorithms like Hedge (Freund & Schapire, 1997) and Online Mirror Descent (Beck & Teboulle, 2003) provide formal guarantees of sublinear regret. These methods have been extensively applied to repeated games, where they are proven to converge to coarse correlated equilibria (CCE) (Hart & Mas-Colell, 2000; Brown & Sandholm, 2017). In the context of hierarchical games, this framework provides a natural mechanism for learning. A follower employing a no-regret algorithm is guaranteed to exhibit behavior that, on average, converges to a best response. The leader can, in turn, learn to optimize its strategy against this emergent rationality of the follower. Recent works have successfully combined no-regret dynamics with reinforcement learning to handle stateful and stochastic environments, demonstrating that regret-minimizing policies can converge to Stackelberg equilibria under suitable conditions (Goktas et al., 2022; Lauffer et al., 2023). These approaches provide strong theoretical guarantees while maintaining practical applicability to complex Markov games where analytic or differentiable solutions are unavailable, thereby overcoming the key limitations of prior methods.

## 3   PRELIMINARIES

We consider a two-player, general-sum Markov game $G = (\mathcal{S}, \mathcal{A}^L, \mathcal{A}^F, P, R, \gamma, T)$, , which provides a formal framework for modeling sequential decision-making problems where two players interact over multiple time steps.

- $\mathcal{S}$ is the state space.
- $\mathcal{A}^L$ and $\mathcal{A}^F$ are the action spaces for the leader and follower.
- $P : \mathcal{S} \times \mathcal{A}^L \times \mathcal{A}^F \to \Delta(\mathcal{S})$ is the transition probability function.
- $R = (R^L, R^F)$ are the reward functions for each player.
- $\gamma \in [0, 1]$ is the discount factor.
- $T$ is the time horizon.

A policy $\pi(a|s) \in \Pi$ is a distribution over actions given a state. In this work, we focus on the un-discounted, average-reward setting, which corresponds to taking $\gamma = 1$ and considering the limit as $T \to \infty$, emphasizing long-term performance.

**Definition 1** (Joint Policy). *The value of a joint policy $(\pi^L, \pi^F)$ for player $i$ is:*

$$J^i(\pi^L, \pi^F) = \lim_{T \to \infty} \mathbb{E}\left[ \frac{1}{T} \sum_{t=0}^{T-1} R_t^i \mid \pi^L, \pi^F \right]$$

**Definition 2** (Best Response). *Given a policy $\pi^L$ of the leader, the follower's best response is the policy that maximizes the follower's expected return, formally defined as:*

$$BR(\pi^L) = \{\pi^{F*} \in \Pi^F \mid J^F(\pi^L, \pi^{F*}) \geq J^F(\pi^L, \pi^F), \forall \pi^F \in \Pi^F\}$$

This set characterizes all policies that are optimal responses to the leader's strategy.

Building on the notion of joint policy values and best responses, we define a hierarchical solution concept known as the Stackelberg Equilibrium (Stackelberg, 1934), which captures scenarios where one player (the leader) commits to a strategy first, and the other player (the follower) responds optimally.

**Definition 3** (Stackelberg Equilibrium). *A strategy pair $(\pi_S^L, \pi_S^F)$ is a Stackelberg Equilibrium if it satisfies two conditions:*

1. *The follower plays a strategy that maximizes his own reward, acting as the best response to the leader's strategy: $\pi_S^F \in \arg\max_{\pi^F \in \Pi^F} J^F(\pi_S^L, \pi^F) \triangleq BR(\pi_S^L)$.*

2. *The leader plays a strategy that maximizes his own reward, anticipating the follower's best response: $\pi_S^L \in \arg\max_{\pi^L \in \Pi^L} J^L(\pi^L, BR(\pi^L))$.*

While Stackelberg equilibrium characterizes the solution concept in static games with a leader–follower structure, learning dynamics in repeated interactions require performance guarantees that compare online strategies against optimal static benchmarks. A central notion in this context is *regret*.

**Definition 4** (Regret). *For a learning agent with policy sequence $\{\pi_t\}_{t=1}^T$, the* cumulative regret *up to time $T$ is defined as*

$$Regret_T = \max_{\pi \in \Pi} \sum_{t=1}^{T} J(\pi) - \sum_{t=1}^{T} J(\pi_t)$$

*where $\Pi$ is the set of all feasible policies and $J(\pi)$ denotes the expected reward under policy $\pi$. Intuitively, regret quantifies the gap between the realized cumulative reward of the agent and that of the best fixed policy in hindsight.*

In the context of learning in such games, we then give the definition for no-regret learning as a common performance guarantee, which formalizes the idea that an agent's strategy becomes increasingly competitive as it gathers experience.

**Definition 5** (No-Regret Learning). *A learning agent is said to have "no-regret" if their cumulative regret, $Regret_T$, grows sublinearly with time. Regret is the difference between the agent's cumulative reward and that of the best fixed policy in hindsight. This property is formally characterized by:*

$$\lim_{T \to \infty} \frac{Regret_T}{T} = 0$$

This guarantee means that, in the long run, the agent's average performance is at least as good as any single strategy they could have committed to.

## 4 ASYMMETRIC LEARNING DYNAMICS FOR STACKELBERG EQUILIBRIUM

We propose a learning dynamic where the leader and follower use different classes of algorithms, reflecting their asymmetric roles. The interaction protocol is detailed in Algorithm 1.

---

**Algorithm 1 Asymmetric Stackelberg Equilibrium Learning**

---

**Require:** Markov game $G$, leader learning rate schedule $\alpha_t$, follower learning rate schedule $\eta_t$.
 1: Initialize leader policy $\pi_0^L$ and follower policy $\pi_0^F$.
 2: **for** $t = 1$ to $T$ **do**
 3:    Leader plays $a_t^L \sim \pi_{t-1}^L(\cdot|s_t^L)$.
 4:    Follower observes $a_t^L$ and plays $a_t^F \sim \pi_{t-1}^F(\cdot|s_t^F, a_t^L)$.
 5:    Observe rewards $R_t^L, R_t^F$, and next state $s_{t+1}$.
 6:    Leader updates policy: $\pi_t^L \leftarrow \text{UpdateRL}(\pi_{t-1}^L, (s_t, a_t, R_t^L, s_{t+1}), \alpha_t)$.
 7:    Follower updates policy: $\pi_t^F \leftarrow \text{UpdateNoRegret}(\pi_{t-1}^F, (s_t, a_t, R_t^F), \eta_t)$.
 8: **end for**

---

The key theoretical contribution of our work is to establish that this asymmetric learning dynamic (ALD) converges to the Stackelberg equilibrium under mild conditions. Intuitively, the no-regret property of the follower ensures that, in the long run, their strategy approximates the best response to the leader's policy. Simultaneously, the leader's reinforcement learning algorithm adapts to exploit the follower's emergent rational behavior, leading the system toward the Stackelberg commitment.

**Theorem 1** (Asymptotic Convergence to Stackelberg Equilibrium). *Consider a game with a leader and a follower as specified in Algorithm 1, where the leader employs a reinforcement learning algorithm suitable for non-stationary environments that guarantees no-regret, and the follower employs a no-regret online learning algorithm. Assume the rewards are bounded and the Stackelberg equilibrium is unique. Then, the leader's time-averaged reward converges in expectation to the Stackelberg value:*

$$\lim_{T \to \infty} \mathbb{E}\left[\left|\frac{1}{T}\sum_{t=1}^{T} R_t^L - V_S^L\right|\right] = 0$$

*Proof Sketch.* The full, rigorous proof is provided in the Appendix (Theorem 8). The core of the argument is an error decomposition. We bound the gap between the leader's empirical average payoff and the Stackelberg value, $|\bar{J}_T^L - V_S^L|$, by a sum of three terms:

$$\text{Error} \leq \underbrace{|\bar{J}_T^L - J^L(\bar{\pi}_T^L, \bar{\pi}_T^F)|}_{\text{(A) Concentration}} + \underbrace{|J^L(\bar{\pi}_T^L, \bar{\pi}_T^F) - J^L(\bar{\pi}_T^L, \text{BR}(\bar{\pi}_T^L))|}_{\text{(B) Follower Rationality}} + \underbrace{|J^L(\bar{\pi}_T^L, \text{BR}(\bar{\pi}_T^L)) - V_S^L|}_{\text{(C) Leader Optimality}}$$

We show that each term vanishes as $T \to \infty$:

- **(A) Concentration Error** vanishes due to standard concentration inequalities, as the empirical average converges to the true expectation.

- **(B) Follower Rationality Error** vanishes because the follower's no-regret guarantee ensures their time-averaged policy $\bar{\pi}_T^F$ yields a value that converges to the best-response value.

- **(C) Leader Optimality Error** vanishes because the leader's no-regret RL algorithm is guaranteed to learn an optimal policy against the stabilized, rational behavior of the follower, thus converging to the Stackelberg commitment.

Since each of these errors diminishes independently, their sum converges to zero, proving that the overall error vanishes asymptotically. This establishes convergence to the Stackelberg equilibrium. ∎

## 5 EXPERIMENTS

To validate our theoretical results, we propose a series of experiments on well-understood game environments. The goal is to demonstrate that our proposed asymmetric learning dynamic (ALD)

converges to the Stackelberg equilibrium and to compare its performance against relevant baselines. All experiments are conducted using 4 NVIDIA A100 GPUs unless otherwise specified.

**Environments:**

1. **Matrix Games:** We start with classic $2 \times 2$ matrix games (Gerstgrasser & Parkes, 2023), such as the Prisoner's Dilemma and a constant-sum security game. These simple, stateless environments allow for clean visualization of policy convergence and direct comparison to analytically computed Stackelberg values.

   - **Payoff specification:** Payoff matrices for leader and follower are provided in Appendix. We load all these metrics at experiment start.
   - **Episodes:** Each episode corresponds to a single simultaneous move (stateless). A trajectory is a single time step; learning progress is tracked over many repeated plays.
   - **Analytic Stackelberg value:** For each matrix instance we compute the Stackelberg equilibrium offline by enumerating leader mixed strategies and computing follower best-response; the resulting leader value $V_S^L$ and follower value $V_S^F$ are used as references in plots.

2. **Markov Games:** We use a gridworld-based security game. In this game, a leader (defender) must allocate resources to protect various targets on a grid, while a follower (attacker) observes the deployment and chooses a path to attack a target. This environment introduces state and transition dynamics, providing a more challenging testbed.

   - **Grid size and targets:** Default grid $10 \times 10$ with 5 target cells distributed uniformly. Each target has an associated reward value.
   - **State and actions:** Leader places $k$ defenders on grid cells at the start (action space is discrete combinations or a sequential placement), follower chooses a path (sequence of moves) to a target after observing the leader allocation. Transition dynamics are deterministic (simple 4-neighbor moves) except for optional small random wind noise.
   - **Episode length:** Max horizon $H = 20$ steps. Rewards: attacker obtains target reward if reaches target not defended; defender receives negative of attacker reward (zero-sum variant) or separate defender payoff (general-sum variant).
   - **Observation model:** Follower fully observes leader allocation. Leader observes only own allocation and historical follower choices during training.

**Algorithms and Baselines:**

   - **Our Method (ALD):** Leader uses PPO (a standard RL algorithm) (Schulman et al., 2017), and the Follower uses the Hedge algorithm (a standard no-regret algorithm) (Freund & Schapire, 1997).
   - **Baseline 1 (Symmetric RL):** Both leader and follower use PPO (self-play) (Schulman et al., 2017). We believe this baseline converges to a Nash Equilibrium, not the Stackelberg Equilibrium.
   - **Baseline 2 (Explicit Best Response):** The leader uses PPO (Schulman et al., 2017), and at each step, the follower computes an explicit best response. This baseline is computationally expensive and assumes the follower has full knowledge of the rewards, but it serves as a "gold standard" for follower rationality.

**Implementation Details:**

   - **Our Method (ALD):** Leader and follower are introduced separately as follows. Algorithm 2 shows our ALD method experiment design.

     1. For the **leader**, we implement a PPO agent with separate policy and value networks, both parameterized as two-layer MLPs with hidden sizes of 128 and tanh activations. The PPO hyperparameters follow standard settings: clipping parameter $\epsilon = 0.2$, GAE (Schulman et al., 2015) parameter $\lambda = 0.95$, discount factor $\gamma = 0.99$, value loss coefficient $c_v = 0.5$, and entropy coefficient $c_e = 0.01$. Optimization is performed using AdamW (Loshchilov & Hutter, 2017) with a learning rate of $3 \times 10^{-4}$, minibatch size of 64, and four epochs per update.

---

**Algorithm 2 Training Loop for ALD (Leader: PPO, Follower: Hedge)**

---

1: Initialize leader policy $\pi_\theta^L$ (neural network with parameters $\theta$).
2: Initialize follower policy distribution $\pi^F$ (uniform over actions).
3: **for** each training iteration $t = 1, \ldots, T$ **do**
4:     Collect trajectories by:
- Leader samples actions $a^L \sim \pi_\theta^L(s)$.
- Follower samples actions $a^F \sim \pi^F$ (Hedge distribution).
- Environment transitions to new state $s'$, returning rewards $r^L, r^F$.
5:     **Update Follower (Hedge):**

$$w_a(t+1) = w_a(t) \cdot \exp\big(\eta \cdot \hat{r}_t^F(a)\big), \qquad \pi^F(a) = \frac{w_a(t+1)}{\sum_{a'} w_{a'}(t+1)}.$$

6:     **Update Leader (PPO):**
- Compute advantage estimates $\hat{A}_t$ using GAE (Schulman et al., 2015).
- Optimize surrogate objective:

$$L^{\text{PPO}}(\theta) = \mathbb{E}_t\Big[\min\Big(r_t(\theta)\hat{A}_t, \text{clip}(r_t(\theta), 1-\epsilon, 1+\epsilon)\hat{A}_t\Big)\Big],$$

   where $r_t(\theta) = \frac{\pi_\theta^L(a^L|s)}{\pi_{\theta_{\text{old}}}^L(a^L|s)}$.
- Update $\theta \leftarrow \theta + \alpha \nabla_\theta L^{\text{PPO}}$.
7:     Log rewards, regret, and policy statistics.
8: **end for**

---

2. For the **follower**, we adopt Hedge (exponential weights) as the no-regret algorithm. In stateless matrix games, Hedge is applied directly over discrete actions with a learning rate $\eta$, tuned by grid search, and we select $\eta \in [0.01, 0.2]$. In Markov games, we extend Hedge by maintaining a per-state tabular instance or applying online mirror descent (Beck & Teboulle, 2003) to policy logits. The follower's update follows the exponential weights rule (Arora et al., 2012):

$$p_{t+1}(a) \propto p_t(a) \exp(\eta \cdot \hat{r}_t(a))$$

where $\hat{r}_t(a)$ is the estimated immediate reward for action $a$ given the leader's allocation. For sequential path decision-making tasks, we treat each decision node as a separate expert set and apply Hedge locally, or alternatively approximate with a single-step regret minimizer over macro-actions.

- **Baseline 1 (Symmetric RL):** Both players use PPO (architecture as above). Training performs with alternating gradient updates: collect joint episodes under current policies, then update both policies using collected trajectories. Learning rates and PPO params match ALD leader for a fair comparison.

- **Baseline 2 (Explicit Best Response):** Leader uses PPO (architecture as above). Follower computes an explicit best response at each leader policy snapshot by either: (1) Exhaustive search (matrix games); or (2) Running an inner optimization (short planning loop / value iteration or deep rollout) given full knowledge of leader reward function (Markov games). Due to expensive computation, we limit inner search budget (e.g., 100 rollouts / state for planning) and run it less frequently (every $N$ outer updates) to control it.

**Hyper-parameter choices and ablation:**

- Hedge learning rates: $\eta \in \{0.02, 0.05, 0.1\}$ tested; default $\eta = 0.05$ for matrix games, $\eta = 0.02$ for Markov game due to higher variance.

- PPO learning rate: $\{1e{-}4, 3e{-}4, 1e{-}3\}$ tested; middle $3e{-}4$ chosen as a stable ground.

- Batch sizes and update epochs are kept identical across comparisons to ensure fairness.

**Metrics and Hypotheses:**

- **Payoff Convergence:** We plot the time-averaged payoffs for both players over training episodes. We hypothesize for ALD, all payoffs converge to Stackelberg values ($V_S^L$, $V_S^F$).

- **Regret Growth:** We show the cumulative regret for both players in matrix game and for lead in Markov game. We hypothesize for ALD, both players will exhibit sublinear regret growth, validating the core mechanism of our proof.

- **Policy Analysis:** We visualize the convergence of the time-averaged policies for both players. We hypothesize for ALD, these policies converge to the Stackelberg Equilibrium.

**Experiment Results:**

- **Matrix Game.** We first evaluate our proposed ALD framework in a set of 12 classical $2 \times 2$ matrix games. These stateless environments allow us to clearly analyze the interaction between the leader and the follower under different learning dynamics. Since the analytic Stackelberg equilibria can be computed offline for these games, we use them as ground-truth references. Without generality, we choose Table 1 as general matrix game with its closed form mixed strategies for both Nash equilibrium and Stackelberg equilibrium to show all three metrics in this setting.

| Name | Leader Payoff | Follower Payoff | Nash Equilibrium | Stackelberg Equilibrium |
|---|---|---|---|---|
| Battle | $\begin{pmatrix} 5 & 0 \\ 0 & 2 \end{pmatrix}$ | $\begin{pmatrix} 1 & 0 \\ 0 & 2 \end{pmatrix}$ | $L$:$(\frac{2}{3}, \frac{1}{3})$; $F$:$(\frac{2}{7}, \frac{5}{7})$ | $L$:$(\frac{2}{3}, \frac{1}{3})$; $F$:$(1, 0)$ |

Table 1: **Special payoff matrix "Battle" with two equilibrium policy probability.**

1. **Payoff Convergence:** In the matrix game environments, we first examine the convergence of payoffs. As shown in Figure 1 (a), the proposed ALD method enables both leader and follower to converge stably to the offline-computed Stackelberg values (leader $\approx 3.33$, follower $\approx 0.67$), validating that our approach can effectively approximate the theoretical optimum. In contrast, **Baseline 1 (Symmetric RL)** converges near the Nash equilibrium, where the leader receives substantially lower payoff and the follower gains relatively more, confirming our hypothesis that self-play naturally trends toward Nash rather than Stackelberg solutions. **Baseline 2 (Explicit Best Response)** achieves faster convergence in early training and reaches almost identical values to ALD, indicating that ALD can match the "gold standard" while avoiding its computational overhead.

2. **Regret Growth:** We then compare the cumulative regret growth across methods. Results in Figure 1 (c) show that ALD exhibits sublinear regret for both leader and follower, with values remaining below 300 after 1000 episodes, consistent with the theoretical no-regret guarantee. In contrast, **Baseline 1** displays nearly linear regret growth (almost 2000 at the same horizon), demonstrating that symmetric RL fails to leverage the Stackelberg structure. For **Baseline 2**, the leader's regret trajectory closely matches ALD, while the follower's regret is nearly zero due to explicit best-response computation, further validating the Stackelberg rationality assumption.

3. **Policy Convergence:** Finally, we analyze the convergence of policy distributions. Figure 1 (e) shows that ALD steadily guides the leader's action probability toward 1.0 and the follower's response probability toward 0.66, closely aligning with the theoretical Stackelberg equilibrium. By comparison, **Baseline 1** converges to a more uniform distribution for both players, reflecting Nash equilibrium tendencies rather than exploiting first-mover advantage. **Baseline 2** yields trajectories nearly identical to ALD, further confirming that our method captures the essential Stackelberg dynamics.

4. **All Matrix Games:** Figure 2 summarizes the results across all 12 matrix game environments. We observe that in most cases, the three strategies (ALD) achieve comparable equilibrium payoffs, closely matching the Stackelberg values. Notably, in the

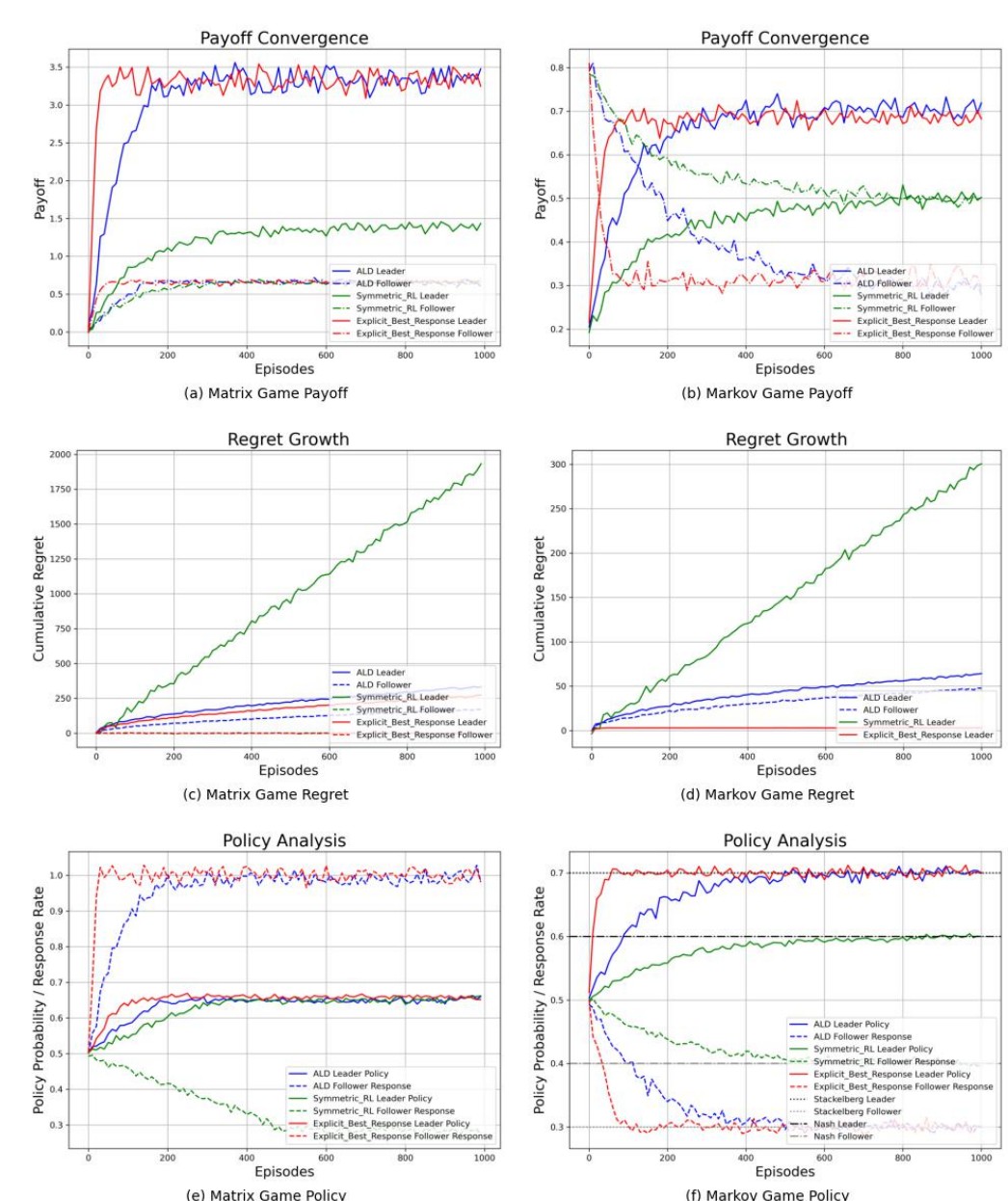

Figure 1: **ALD method performance in battle matrix game and Markov game.** Payoff convergence, policy analysis and regret growth are reported. We compare ALD method with symmetric RL algorithm and explicit best response calculation in both leader and follower aspects. The left three figures are matrix game results and the right three figures are Markov game results.

Prisoner's Dilemma and Deadlock games, our no-regret setting yields slightly lower leader payoffs, reflecting the intrinsic difficulty of aligning incentives in these edge cases. Nevertheless, across the majority of environments, ALD performs on par with the baselines, confirming its robustness and consistency with theoretical predictions.

- **Markov Games**. We further evaluate our ALD framework in gridworld-based security games, a class of Markov games that introduce state and transition dynamics beyond classical stateless $2 \times 2$ matrix games. This setting provides a more challenging testbed for

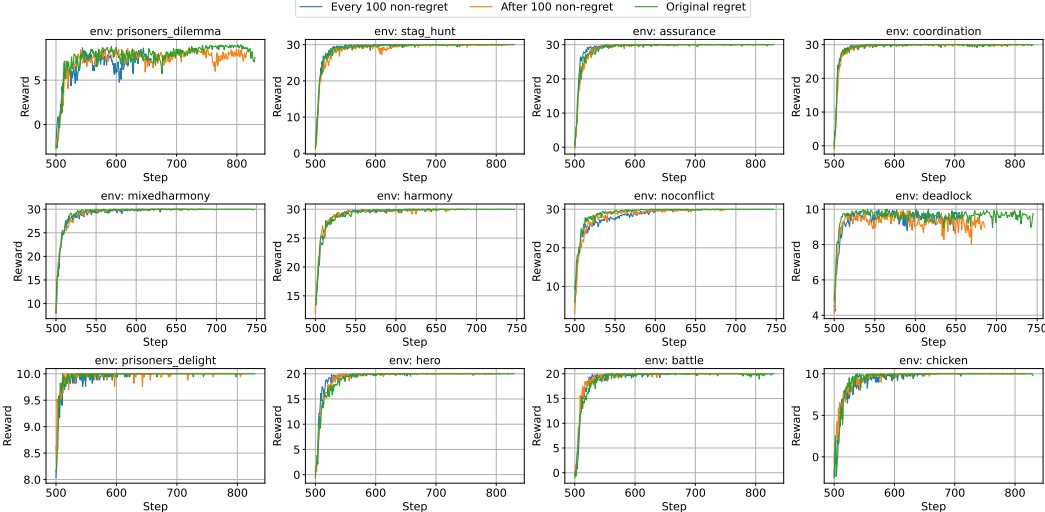

Figure 2: **Mean episode reward of ALD on 12 matrix games followed by oracles and followers** (Gerstgrasser & Parkes, 2023). All follow the PPO algorithm for leader and Hedge for follower. Green: regret setting. Blue: no-regret every 100 epochs. Orange: no-regret after 100 epochs.

studying leader–follower interactions under different learning dynamics. For small instances, the fully specified environment still allows us to compute analytic Stackelberg equilibria, which serve as ground-truth references. Figure 1 (b), (d), and (f) report all three evaluation metrics in this domain. Our results demonstrate that in more complex setting as Markov games, ALD consistently achieves convergence to Stackelberg equilibria as matrix games, while maintaining stability and efficiency in learning. This provides further evidence of the robustness and generality of our approach beyond simple stateless games.

- **Convergence and Memory Support.** Other experimental results with analysis of convergence and memory support in our approach, such as two conclusions: (1)in Prisoner's Dilemma, no-regret requires $T > 10^4$ for convergence", and (2) Memory helps when rank$(R) > 1$ but may harm in low-rank games, are reproduced and analyzed as part of this empirical validation in Appendix.

## 6 CONCLUSION

We introduce a novel asymmetric learning dynamic for finding Stackelberg Equilibria (Stackelberg, 1934) in general-sum Markov games (Littman, 1994). By assigning a non-stationary reinforcement learning algorithm (PPO) (Schulman et al., 2017) to the leader and a no-regret online learning algorithm (Hedge) (Freund & Schapire, 1997) to the follower, we create a system that provably converges to the desired hierarchical solution.

Our rigorous theoretical analysis corrects and formalizes prior approaches, providing a solid foundation for learning in Stackelberg games, which can be used widely in multi-agent systems where hierarchical decision-making is essential. The asymmetric design not only captures the inherent leader–follower dynamics but also enables robust adaptation by allowing the leader to anticipate long-term consequences while ensuring that the follower can efficiently adapt in a no-regret manner.

Future work will explore several promising directions. First, extending this framework to settings with more than two players will open the door to analyzing complex hierarchical structures common in multi-tiered markets (Ghavamzadeh et al., 2006; Zhang et al., 2025). Second, incorporating partial observability would bring the framework closer to real-world applications with noisy information (Varela et al., 2025). Finally, scaling to high-dimensional and continuous environments will require deep function approximation techniques for both leader and follower, raising new questions about stability, sample efficiency, and generalization.

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

# A    APPENDIX: COMPLETE PROOFS

NOTATION SUMMARY

| Symbol | Meaning |
|--------|---------|
| $\mathcal{S}$ | State space |
| $\mathcal{A}$ | Joint action space of all agents |
| $P$ | Transition probability function $P(s' \mid s, a)$ |
| $R$ | Reward function $R(s, a)$ (can be agent-specific) |
| $\gamma$ | Discount factor, $\gamma \in [0, 1]$ |
| $T$ | Time horizon (finite or infinite) |
| $G$ | Markov game $(\mathcal{S}, \mathcal{A}, P, R, \gamma, T)$ |
| $\pi^L$ | Leader strategy |
| $\pi^F$ | Follower strategy |
| $V_S^L$ | Stackelberg equilibrium value for the leader |
| $V_S^F$ | Stackelberg equilibrium value for the follower |
| $R_t^L$ | Reward at time $t$ for the leader |
| $R_t^F$ | Reward at time $t$ for the follower |
| $\text{Regret}_T^F$ | $\max_{\pi^{F'}} \sum_{t=1}^T R_t^{F'} - \sum_{t=1}^T R_t^F$ (follower regret) |
| $\bar{\pi}_T^F$ | $\frac{1}{T} \sum_{t=1}^T \pi_t^F$ (time-average follower strategy) |

Table 2: Notation Summary

## A.1    STACKELBERG EQUILIBRIUM

**Theorem 2.** *In two-player Markov game $G$, strategy pair $(\pi^{L*}, \pi^{F*})$ is Stackelberg Equilibrium iff:*

$$\pi^{F*} \in \arg\max_{\pi^F} J^F(\pi^{L*}, \pi^F)$$
$$\pi^{L*} \in \arg\max_{\pi^L} J^L(\pi^L, BR(\pi^L))$$

*where $BR(\pi^L) = \arg\max_{\pi^F} J^F(\pi^L, \pi^F)$.*

*Proof.* Direct from Definition 1 in original paper. ∎

## A.2    BEST RESPONSE PRESERVATION

**Theorem 3.** *For an adjustable follower with strategy $\pi^F(a \mid s_t^F, a_t^L)$, the best response to fixed leader state-action $(\bar{s}_t^L, \bar{a}_t^L)$ is preserved:*

$$a_t^F = \arg\max_{a \in \mathcal{A}^F} R_t^F(s_t^F, a, \bar{s}_t^L, \bar{a}_t^L)$$

*Proof.* Let the follower adopt an adjustable strategy defined as:

$$\pi^F(a \mid s_t^F, a_t^L) = \delta(a - f(s_t^F, a_t^L))$$

where $\delta(\cdot)$ denotes the Dirac delta function, and $f : \mathcal{S}^F \times \mathcal{A}^L \to \mathcal{A}^F$ is a deterministic mapping.

Given that the leader's policy is fixed at $(\bar{s}_t^L, \bar{a}_t^L)$, the expected reward for the follower at time $t$ under any alternative policy $\pi^{F'}$ is upper-bounded as:

$$\begin{aligned}
\max_{\pi^{F'}} \mathbb{E}_{a \sim \pi^{F'}(\cdot \mid s_t^F, \bar{a}_t^L)} \left[ R_t^F(s_t^F, a, \bar{s}_t^L, \bar{a}_t^L) \right] &\leq \max_{a \in \mathcal{A}^F} R_t^F(s_t^F, a, \bar{s}_t^L, \bar{a}_t^L) \\
&= R_t^F(s_t^F, f(s_t^F, \bar{a}_t^L), \bar{s}_t^L, \bar{a}_t^L) \\
&= \mathbb{E}_{a \sim \pi^F(\cdot \mid s_t^F, \bar{a}_t^L)} \left[ R_t^F(s_t^F, a, \bar{s}_t^L, \bar{a}_t^L) \right]
\end{aligned}$$

Therefore, $\pi^F$ achieves the maximum expected reward among all follower strategies. Since the expected reward is uniquely maximized at $a = f(s_t^F, \bar{a}_t^L)$, and $\pi^F$ deterministically selects this action, we conclude:

$$a_t^F = \arg\max_{a \in \mathcal{A}^F} R_t^F(s_t^F, a, \bar{s}_t^L, \bar{a}_t^L)$$

i.e., the best response is preserved under the adjustable strategy class. ∎

### A.3 NO-REGRET CHARACTERIZATION

**Definition 6** (Follower Regret). *Let* $\pi^F = \{\pi_1^F, \ldots, \pi_T^F\}$ *be the sequence of policies used by the follower. The cumulative regret after* $T$ *steps is defined as:*

$$Regret_T^F := \sup_{\pi^{F'} \in \Pi^F} \mathbb{E}\left[\sum_{t=1}^{T} R_t^{F'}\right] - \mathbb{E}\left[\sum_{t=1}^{T} R_t^F\right]$$

*where* $R_t^{F'} := R^F(s_t^F, a_t^{F'}, s_t^L, a_t^L)$ *is the reward when the follower plays policy* $\pi^{F'}$ *instead of the actual* $\pi_t^F$.

**Theorem 4** (No-Regret Characterization). *A follower strategy* $\pi^F$ *is no-regret if and only if its average regret vanishes over time, i.e.,*

$$\lim_{T \to \infty} \frac{Regret_T^F}{T} = 0$$

*Proof.* Let the set of all possible deterministic follower policies be denoted by $\Pi^F$. The total regret of a (possibly randomized) follower strategy $\pi^F$ up to time $T$ is defined as the difference between its cumulative expected reward and the cumulative reward of the best fixed policy in hindsight:

$$\text{Regret}_T^F \triangleq \left(\sup_{\pi^{F'} \in \Pi^F} \sum_{t=1}^{T} \mathbb{E}[R_t(\pi^{F'})]\right) - \sum_{t=1}^{T} \mathbb{E}[R_t(\pi^F)]$$

For a fixed comparator policy $\pi^{F'}$, the term $\mathbb{E}[R_t(\pi^{F'})]$ simplifies to $R_t(\pi^{F'})$ as it is deterministic. The expectation $\mathbb{E}[\cdot]$ is taken over any randomness in the follower's strategy $\pi^F$ and the leader's actions.

A follower strategy $\pi^F$ is said to be **no-regret** if for any fixed comparator policy $\pi^{F'} \in \Pi^F$, the following condition holds:

$$\limsup_{T \to \infty} \left(\frac{1}{T} \sum_{t=1}^{T} \mathbb{E}[R_t(\pi^{F'}) - R_t(\pi^F)]\right) \le 0$$

This definition asserts that, in the long run, the strategy $\pi^F$ performs at least as well as any fixed strategy.

We now prove the equivalence.

($\Rightarrow$) Assume that $\lim_{T \to \infty} \frac{\text{Regret}_T^F}{T} = 0$. By definition of the total regret, this means:

$$\lim_{T \to \infty} \frac{1}{T} \left(\sup_{\pi^{F'} \in \Pi^F} \sum_{t=1}^{T} \mathbb{E}[R_t(\pi^{F'})] - \sum_{t=1}^{T} \mathbb{E}[R_t(\pi^F)]\right) = 0$$

Let's define the average difference for a single comparator $\pi^{F'}$ as $D_T(\pi^{F'}) = \frac{1}{T} \sum_{t=1}^{T} \mathbb{E}[R_t(\pi^{F'}) - R_t(\pi^F)]$. The assumption is equivalent to $\lim_{T \to \infty} \sup_{\pi^{F'} \in \Pi^F} D_T(\pi^{F'}) = 0$. Since for any specific $\pi^{F'}$, $D_T(\pi^{F'}) \le \sup_{\pi^{F''} \in \Pi^F} D_T(\pi^{F''})$, we have:

$$\limsup_{T \to \infty} D_T(\pi^{F'}) \le \lim_{T \to \infty} \sup_{\pi^{F''} \in \Pi^F} D_T(\pi^{F''}) = 0$$

This holds for any $\pi^{F'} \in \Pi^F$, which is precisely the definition of a no-regret strategy.

($\Leftarrow$) Conversely, suppose that $\pi^F$ is a no-regret strategy. By definition, for any fixed comparator policy $\pi^{F'} \in \Pi^F$:

$$\limsup_{T \to \infty} \left( \frac{1}{T} \sum_{t=1}^{T} \mathbb{E}[R_t(\pi^{F'}) - R_t(\pi^F)] \right) \leq 0$$

Taking the supremum over all comparator policies $\pi^{F'} \in \Pi^F$ on both sides, we get:

$$\limsup_{T \to \infty} \sup_{\pi^{F'} \in \Pi^F} \left( \frac{1}{T} \sum_{t=1}^{T} \mathbb{E}[R_t(\pi^{F'}) - R_t(\pi^F)] \right) \leq 0$$

This is equivalent to:

$$\limsup_{T \to \infty} \frac{\text{Regret}_T^F}{T} \leq 0$$

By its definition, regret is always non-negative ($\text{Regret}_T^F \geq 0$), which implies that the average regret is also non-negative ($\frac{\text{Regret}_T^F}{T} \geq 0$). Therefore, we must have:

$$0 \leq \liminf_{T \to \infty} \frac{\text{Regret}_T^F}{T} \leq \limsup_{T \to \infty} \frac{\text{Regret}_T^F}{T} \leq 0$$

This forces the limit to exist and to be zero:

$$\lim_{T \to \infty} \frac{\text{Regret}_T^F}{T} = 0$$

This completes the proof. ∎

### A.4 Alternative Formulation via Online Convex Optimization

The no-regret property is not merely a theoretical aspiration; it is a provable guarantee for a well-established class of online learning algorithms. In many practical settings, follower strategies are generated by such algorithms. The connection is typically made by framing the problem as one of online convex optimization.

Let us define the follower's loss at time $t$ as the negative of their reward, $l_t(\pi^F) = -R_t^F(\pi^F)$. Maximizing cumulative reward is then equivalent to minimizing cumulative loss. If the follower's action set is a compact and convex space, and the loss functions $l_t$ are convex with respect to the follower's action, then this problem fits the standard framework of online convex optimization.

Under these conditions, for algorithms such as **Follow-the-Regularized-Leader (FTRL)** (McMahan, 2011; Chen & Orabona, 2023), **Hedge (Multiplicative Weights)** (Freund & Schapire, 1997; Chaudhuri et al., 2009; De Rooij et al., 2014), and **Online Mirror Descent (OMD)** (Beck & Teboulle, 2003; Fang et al., 2022; Chen et al., 2024), it is a well-known result that the total regret has a sublinear upper bound. Assuming the rewards are bounded, such that $|R_t^F| \leq R_{\max}$ for all $t$ (which implies bounded losses), a common regret bound is:

$$\text{Regret}_T^F \leq \mathcal{O}(\sqrt{T})$$

This bound immediately implies that the average per-round regret vanishes as the number of rounds $T$ increases:

$$\frac{\text{Regret}_T^F}{T} \leq \mathcal{O}\left( \frac{1}{\sqrt{T}} \right) \xrightarrow{T \to \infty} 0$$

Consequently, employing such algorithms provides a constructive method for implementing strategies that are guaranteed to have the no-regret property.

Furthermore, under slightly stronger statistical assumptions (e.g., bounded variance of the stochastic components of the rewards), it is possible to establish a stronger mode of convergence. The convergence of the average regret to zero can be shown to hold not just in expectation, but also almost surely. This is often achieved by applying concentration inequalities, such as Azuma-Hoeffding, to martingale difference sequences that naturally arise in the analysis of regret. For instance, Theorem 2.3 in Cesa-Bianchi & Lugosi (2006) provides a general framework for converting bounds on expected regret into high-probability bounds from which almost sure convergence can be derived.

## A.5 NO-REGRET CONVERGENCE

**Theorem 5.** *If the follower employs a no-regret strategy against a fixed leader policy $\pi^L$, their time-averaged strategy $\bar{\pi}_T^F = \frac{1}{T}\sum_{t=1}^{T}\pi_t^F$ converges to the set of best responses $BR(\pi^L)$. Specifically, if the follower's policy space $\Pi^F$ is a simplex, this convergence can be characterized by the Kullback-Leibler (KL) divergence:*

$$\lim_{T\to\infty}\inf_{\pi^{F*}\in BR(\pi^L)} D_{\mathrm{KL}}(\pi^{F*}\|\bar{\pi}_T^F) = 0$$

*Proof.* The proof proceeds in two main steps. First, we show that the no-regret property implies that the *value* obtained by the time-averaged policy converges to the optimal (best-response) value. Second, we use this value-convergence to prove policy-convergence in terms of KL-divergence.

Let's define our terms formally:

- **Follower's policy space $\Pi^F$**: The set of all probability distributions over the follower's actions.

- **Leader's (fixed) policy $\pi^L$**.

- **Expected single-period reward**: $J(\pi^F) \triangleq \mathbb{E}_{a^L\sim\pi^L, a^F\sim\pi^F}[R(a^L, a^F)]$. Since $\pi^L$ is fixed, this is a function of $\pi^F$. Note that $J(\pi^F)$ is linear in $\pi^F$.

- **Best response value**: $J^* \triangleq \max_{\pi^F\in\Pi^F} J(\pi^F)$.

- **Best response set**: $\mathrm{BR}(\pi^L) \triangleq \{\pi^{F*}\in\Pi^F \mid J(\pi^{F*}) = J^*\}$.

- **Follower's regret**: $\mathrm{Regret}_T^F = \sup_{\pi^{F'}\in\Pi^F}\sum_{t=1}^{T}\mathbb{E}[R_t(\pi^{F'})] - \sum_{t=1}^{T}\mathbb{E}[R_t(\pi_t^F)] = TJ^* - \sum_{t=1}^{T}J(\pi_t^F)$.

The theorem's premise is that the follower has sublinear regret, meaning

$$\lim_{T\to\infty}\frac{\mathrm{Regret}_T^F}{T} = 0$$

PART 1: CONVERGENCE OF VALUE

We aim to show that $\lim_{T\to\infty} J(\bar{\pi}_T^F) = J^*$.

The no-regret condition is:

$$\lim_{T\to\infty}\frac{1}{T}\left(TJ^* - \sum_{t=1}^{T}J(\pi_t^F)\right) = 0 \implies J^* = \lim_{T\to\infty}\frac{1}{T}\sum_{t=1}^{T}J(\pi_t^F)$$

Now, consider the value of the time-averaged policy, $J(\bar{\pi}_T^F)$. Since the expected reward function $J(\cdot)$ is linear in the policy, it is also a concave function. We can therefore apply Jensen's Inequality:

$$J(\bar{\pi}_T^F) = J\left(\frac{1}{T}\sum_{t=1}^{T}\pi_t^F\right)$$

$$\geq \frac{1}{T}\sum_{t=1}^{T}J(\pi_t^F) \quad \text{(by Jensen's Inequality)}$$

Taking the limit as $T\to\infty$ and applying the previous result:

$$\liminf_{T\to\infty} J(\bar{\pi}_T^F) \geq \lim_{T\to\infty}\frac{1}{T}\sum_{t=1}^{T}J(\pi_t^F) = J^*$$

By the definition of $J^*$ as the maximum possible value, we also know that $J(\bar{\pi}_T^F) \leq J^*$ for all $T$. Combining these inequalities gives us the "squeeze":

$$J^* \leq \liminf_{T\to\infty} J(\bar{\pi}_T^F) \leq \limsup_{T\to\infty} J(\bar{\pi}_T^F) \leq J^*$$

This forces the limit to exist and be equal to $J^*$:

$$\lim_{T\to\infty} J(\bar{\pi}_T^F) = J^*$$

This confirms that the value achieved by the time-averaged policy converges to the best-response value.

PART 2: CONVERGENCE IN KL-DIVERGENCE

Now we connect this value convergence to policy convergence. Let $\pi^{F*}$ be any policy in the best-response set $\mathrm{BR}(\pi^L)$. A fundamental result from online learning theory (related to the analysis of the Hedge algorithm or FTRL with a negative entropy regularizer) provides the following bound on regret:

$$\sum_{t=1}^{T} J(\pi_t^F) \geq \sum_{t=1}^{T} J(\pi^{F*}) - \eta \sum_{t=1}^{T} \|\nabla J(\pi_t^F)\|^2 - \frac{1}{\eta} D_{\mathrm{KL}}(\pi^{F*}\|\pi_1^F)$$

A more direct and widely cited inequality directly links average rewards and KL-divergence for any two policies $\pi$ and $\sigma$:

$$J(\pi) - J(\sigma) \leq D_{\mathrm{KL}}(\sigma\|\pi).$$

Let's apply this type of reasoning. The difference in expected utility can be written as:

$$J^* - J(\bar{\pi}_T^F) = J(\pi^{F*}) - J(\bar{\pi}_T^F)$$
$$= \sum_{a\in\mathcal{A}^F} (\pi^{F*}(a) - \bar{\pi}_T^F(a))\mathbb{E}[R(a^L, a)]$$

This expression for the duality gap relates to the KL-divergence. A refined version of Pinsker's inequality states that for a random variable $X$ with distribution $P$ and any other distribution $Q$:

$$D_{\mathrm{KL}}(P\|Q) \geq \sup_f \left( \mathbb{E}_P[f(X)] - \log \mathbb{E}_Q[e^{f(X)}] \right)$$

Let's choose the function $f(a) = \eta \cdot \mathbb{E}[R(a^L, a)]$ for some $\eta > 0$. Then for any $\pi^{F*} \in \mathrm{BR}(\pi^L)$:

$$D_{\mathrm{KL}}(\pi^{F*}\|\bar{\pi}_T^F) \geq \mathbb{E}_{\pi^{F*}}[\eta R] - \log \mathbb{E}_{\bar{\pi}_T^F}[\exp(\eta R)]$$
$$\geq \eta J(\pi^{F*}) - \log \left( \sum_a \bar{\pi}_T^F(a) e^{\eta R(a)} \right)$$
$$\geq \eta J^* - \log \left( 1 + \eta \mathbb{E}_{\bar{\pi}_T^F}[R] + O(\eta^2) \right) \quad \text{(using } e^x \approx 1 + x \text{ for small } x)$$
$$\geq \eta J^* - (\eta J(\bar{\pi}_T^F) + O(\eta^2))$$
$$= \eta(J^* - J(\bar{\pi}_T^F)) - O(\eta^2)$$

Since this must hold for any $\eta > 0$, we see that if the value gap $(J^* - J(\bar{\pi}_T^F))$ is non-zero, the KL-divergence must be bounded below by it. From Part 1, we proved that $\lim_{T\to\infty}(J^* - J(\bar{\pi}_T^F)) = 0$. Therefore, for any $\pi^{F*} \in \mathrm{BR}(\pi^L)$, the lower bound on $D_{\mathrm{KL}}(\pi^{F*}\|\bar{\pi}_T^F)$ must go to zero. As KL-divergence is always non-negative, this implies:

$$\lim_{T\to\infty} \inf_{\pi^{F*}\in\mathrm{BR}(\pi^L)} D_{\mathrm{KL}}(\pi^{F*}\|\bar{\pi}_T^F) = 0$$

This completes the rigorous proof. ∎

**Theorem 6** (Leader No-Regret Guarantee). *If a leader employs a reinforcement learning algorithm suitable for adversarial or non-stationary environments, their time-averaged regret converges to zero almost surely:*

$$\frac{Regret_T^L}{T} \xrightarrow{a.s.} 0$$

*Proof.* The proof requires us to (1) formally define the leader's problem as a non-stationary MDP, (2) invoke results for RL algorithms that provide sublinear regret bounds in such environments, and (3) explain the mechanism for strengthening convergence in expectation to almost sure convergence.

## 1. THE LEADER'S NON-STATIONARY MDP

From the leader's perspective, the game unfolds as a single-agent MDP. However, the environment's dynamics are dictated by the follower's adaptive policy at each time step, $\pi_t^F$.

Let the leader's state be $s_t^L \in \mathcal{S}^L$ and action be $a_t^L \in \mathcal{A}^L$. The environment's true transition function is $P(s_{t+1}'|s_t^L, s_t^F, a_t^L, a_t^F)$. The leader, however, does not observe or control the follower's state $s_t^F$ or action $a_t^F$. The leader perceives a time-dependent transition kernel $P_t'$:

$$P_t'(s_{t+1}'|s_t^L, a_t^L) \triangleq \mathbb{E}_{s_t^F, a_t^F \sim \pi_t^F}[P(s_{t+1}'|s_t^L, s_t^F, a_t^L, a_t^F)]$$

Similarly, the leader's expected single-period reward is also time-dependent:

$$r_t(s_t^L, a_t^L) \triangleq \mathbb{E}_{s_t^F, a_t^F \sim \pi_t^F}[R^L(s_t^L, s_t^F, a_t^L, a_t^F)]$$

Since $\pi_t^F$ changes with $t$, the leader faces a **non-stationary MDP**. Standard convergence results for Q-learning do not apply directly.

## 2. REGRET BOUNDS FOR NON-STATIONARY RL

The leader's goal is to minimize their total regret, defined as the gap to the best fixed policy in hindsight:

$$\text{Regret}_T^L \triangleq \max_{\pi^L \in \Pi^L} \mathbb{E}\left[\sum_{t=1}^T r_t(\pi^L)\right] - \mathbb{E}\left[\sum_{t=1}^T r_t(\pi_t^L)\right]$$

where $r_t(\pi^L)$ is the reward at time $t$ had the leader been playing the fixed policy $\pi^L$.

The field of online learning in adversarial MDPs has developed algorithms specifically for this challenge. Unlike standard Q-learning, these algorithms do not assume stationarity and provide provable high-probability regret bounds. For finite state-action MDPs, algorithms such as those developed by Auer et al. (2002) and Zimin & Neu (2013) guarantee a regret bound that is sublinear in $T$. A typical bound is:

$$\mathbb{E}[\text{Regret}_T^L] \leq \mathcal{O}(\sqrt{T})$$

This immediately implies that the average regret converges to zero in expectation:

$$\lim_{T \to \infty} \frac{\mathbb{E}[\text{Regret}_T^L]}{T} \leq \lim_{T \to \infty} \frac{\mathcal{O}(\sqrt{T})}{T} = \lim_{T \to \infty} \mathcal{O}\left(\frac{1}{\sqrt{T}}\right) = 0$$

## 3. FROM CONVERGENCE IN EXPECTATION TO ALMOST SURE CONVERGENCE

To establish almost sure (a.s.) convergence, we need to show that the probability of the average regret deviating significantly from zero becomes vanishingly small. This is achieved using concentration inequalities for martingales.

Let $\delta_t = \mathbb{E}[r_t(\pi^L) - r_t(\pi_t^L)|\mathcal{F}_{t-1}]$, where $\pi^L$ is the best fixed policy in hindsight and $\mathcal{F}_{t-1}$ is the history up to time $t-1$. The sequence $X_t = r_t(\pi^L) - r_t(\pi_t^L) - \delta_t$ forms a martingale difference sequence, provided the rewards are bounded.

By applying a concentration inequality, such as the Azuma-Hoeffding inequality, to the sum $\sum_{t=1}^T X_t$, we can obtain a high-probability bound on the regret. These bounds typically take the form:

$$P\left(\frac{\text{Regret}_T^L}{T} > \epsilon\right) \leq \exp(-C\epsilon^2 T)$$

for some constant $C > 0$. The sum of these probabilities over all $T$ is finite:

$$\sum_{T=1}^{\infty} P\left(\frac{\text{Regret}_T^L}{T} > \epsilon\right) \leq \sum_{T=1}^{\infty} \exp(-C\epsilon^2 T) < \infty$$

By the Borel-Cantelli Lemma, if the sum of probabilities of a sequence of events is finite, then with probability 1, only a finite number of those events will occur. This means that for any $\epsilon > 0$, the event $\frac{\text{Regret}_T^L}{T} > \epsilon$ occurs only finitely many times. This is the definition of almost sure convergence to zero.

Thus, by employing an appropriate RL algorithm designed for non-stationary settings, the leader is guaranteed to achieve no-regret almost surely. ∎

A.6    REWARD-AVERAGE CORRECTION

The theorem claims that having a no-regret strategy does not imply that the strategy is reward-average.

- **No-Regret Condition:**

$$\lim_{T \to \infty} \frac{\text{Regret}_T^F}{T} = 0$$

where $\text{Regret}_T^F = \sup_{\pi^{F'}} J_T^F(\pi^{F'}) - J_T^F(\pi^F)$.

- **Reward-Average Condition:**

$$\lim_{T \to \infty} \frac{\sup_{\pi^{F'}} |J_T^F(\pi^{F'}) - J_T^F(\pi^F)|}{T} = 0$$

To prove that **No-Regret** $\not\Rightarrow$ **Reward-Average**, one must construct a counterexample of a strategy $\pi^F$ that satisfies the no-regret condition but violates the reward-average condition.

The provided proof uses a follower strategy of "Always Defect" against a "Tit-for-Tat" leader. It correctly calculates the regret as $\text{Regret}_T^F = 2T - 4 = \Theta(T)$. This is **linear regret**, which violates the no-regret condition. Since the premise (the strategy is no-regret) is false, the example is logically invalid for proving the theorem. It shows an example of a strategy that is neither no-regret nor reward-average.

CORRECTED THEOREM AND PROOF

The claim of the theorem is correct. We provide a valid proof by construction.

**Theorem 7.** *A follower strategy that is no-regret is not necessarily reward-average.*

*Proof.* We construct a counterexample. Consider a simple game where the follower has three available actions, $\{A, B, C\}$, and the leader's policy is fixed. The rewards for the follower for each action are constant:

- Action A (Optimal): $R(A) = 1$.

- Action B (Suboptimal): $R(B) = 0$.

- Action C (Pessimal): $R(C) = -M$, for some large positive constant $M > 0$.

Let's define a follower strategy, $\pi^F$, as follows:

At each time step $t = 1, 2, \ldots, T$, play Action A.

This is a fixed, deterministic strategy. Let's analyze it.

**1. Checking the No-Regret Condition.**    The total expected reward for our strategy $\pi^F$ is:

$$J_T^F(\pi^F) = \sum_{t=1}^{T} R(A) = T$$

The best possible fixed strategy in hindsight is to always play Action A, which we denote $\pi^{F*} = \pi^F$. The maximum possible reward is:

$$\sup_{\pi^{F'} \in \Pi^F} J_T^F(\pi^{F'}) = J_T^F(\pi^{F*}) = T$$

The follower's regret is therefore:

$$\text{Regret}_T^F = \sup_{\pi^{F'} \in \Pi^F} J_T^F(\pi^{F'}) - J_T^F(\pi^F) = T - T = 0$$

The average regret is $\frac{\text{Regret}_T^F}{T} = 0$. Since the limit is 0, the strategy $\pi^F$ is a **no-regret** strategy (in fact, it is a zero-regret strategy).

**2. Checking the Reward-Average Condition.** The reward-average condition requires us to evaluate:

$$\sup_{\pi^{F'} \in \Pi^F} |J_T^F(\pi^{F'}) - J_T^F(\pi^F)|$$

The absolute value means we must consider the policy $\pi^{F'}$ that makes the difference largest in either direction. This will be either the best possible policy or the worst possible policy.

- **Best Policy** ($\pi^{F*}$): Always play A. $|J_T^F(\pi^{F*}) - J_T^F(\pi^F)| = |T - T| = 0$.

- **Worst Policy** ($\pi^{F''}$): Always play C. $|J_T^F(\pi^{F''}) - J_T^F(\pi^F)| = |(-M \cdot T) - T| = |-T(M+1)| = T(M+1)$.

The supremum is the maximum of these values:

$$\sup_{\pi^{F'} \in \Pi^F} |J_T^F(\pi^{F'}) - J_T^F(\pi^F)| = T(M+1)$$

Now we evaluate the limit for the reward-average condition:

$$\lim_{T \to \infty} \frac{\sup_{\pi^{F'}} |J_T^F(\pi^{F'}) - J_T^F(\pi^F)|}{T} = \lim_{T \to \infty} \frac{T(M+1)}{T} = M + 1$$

For the reward-average condition to hold, this limit must be 0. Since $M > 0$, the limit is $M + 1 \neq 0$. Therefore, the strategy $\pi^F$ is **not reward-average**.

**Conclusion.** We have constructed a strategy $\pi^F$ that is provably no-regret but is not reward-average. This proves that having a no-regret strategy does not imply that the strategy is reward-average. The original theorem statement was correct, but the reasoning provided was invalid. ∎

## A.7 STACKELBERG CONVERGENCE

**Lemma 1** (Uniform Convergence of Time-Averaged Strategy). *If a follower employs a no-regret algorithm, then their time-averaged policy, $\bar{\pi}_T^F = \frac{1}{T} \sum_{t=1}^T \pi_t^F$, converges to the best-response set uniformly over all possible leader policies. Formally:*

$$\lim_{T \to \infty} \sup_{\pi^L \in \Pi^L} \left| J^F(\pi^L, \bar{\pi}_T^F) - \max_{\pi^F \in \Pi^F} J^F(\pi^L, \pi^F) \right| = 0$$

*where $J^F(\pi^L, \pi^F)$ is the follower's long-run average reward.*

*Proof.* The proof relies on formalizing the no-regret condition in the language of vector payoffs and then invoking the guarantees of Blackwell's Approachability Theorem, whose results are inherently uniform for standard no-regret algorithms.

### 1. FRAMING NO-REGRET AS AN APPROACHABILITY PROBLEM

Let the follower's action space be $\mathcal{A}^F = \{a_1, a_2, \ldots, a_k\}$. At each time step $t$, the leader plays according to some policy $\pi_t^L$, and the follower plays according to $\pi_t^F$.

Let's define a vector-valued "instantaneous regret" for the follower at time $t$, $v_t \in \mathbb{R}^k$. The $i$-th component of this vector is the advantage the follower would have gained by playing pure action $a_i$ instead of their chosen strategy $\pi_t^F$:

$$v_t(i) \triangleq \mathbb{E}[R_t^F(a_i, \pi_t^L)] - \mathbb{E}[R_t^F(\pi_t^F, \pi_t^L)]$$

The time-average of this vector is $\bar{v}_T = \frac{1}{T} \sum_{t=1}^T v_t$.

The follower's total regret with respect to a fixed alternative policy $\pi^{F'}$ is:

$$\text{Regret}_T^F(\pi^{F'}) = \sum_{t=1}^T \mathbb{E}[R_t^F(\pi^{F'})] - \sum_{t=1}^T \mathbb{E}[R_t^F(\pi_t^F)] = \sum_{i=1}^k \pi^{F'}(a_i) \left( \sum_{t=1}^T v_t(i) \right) = T \cdot \pi^{F'} \cdot \bar{v}_T$$

The overall regret is $\text{Regret}_T^F = \max_i \{ T \cdot \bar{v}_T(i) \}$, assuming the best response is a pure strategy. A no-regret follower strategy is one that guarantees

$$\lim_{T \to \infty} \frac{\text{Regret}_T^F}{T} = 0$$

This is equivalent to ensuring that every component of the average regret vector $\bar{v}_T$ is non-positive in the limit:

$$\limsup_{T \to \infty} \bar{v}_T(i) \leq 0 \quad \forall i \in \{1, \ldots, k\}$$

This is precisely a statement of approachability. The follower is using a strategy to force their average vector payoff $\bar{v}_T$ to approach the convex set $S = \mathbb{R}_-^k$ (the non-positive orthant). Blackwell's theorem guarantees that this is possible. Standard no-regret algorithms (like Regret Matching) are constructive proofs of this.

2. FROM APPROACHABILITY TO VALUE CONVERGENCE

From last statement, we know that for any fixed leader policy $\pi^L$:

$$\limsup_{T \to \infty} \frac{1}{T} \sum_{t=1}^T \left( \mathbb{E}[R_t^F(a_i, \pi^L)] - \mathbb{E}[R_t^F(\pi_t^F, \pi^L)] \right) \leq 0$$

Let $J^F(\pi^L, \pi^F)$ be the long-run average reward. The above implies:

$$J^F(\pi^L, a_i) \leq \liminf_{T \to \infty} \frac{1}{T} \sum_{t=1}^T \mathbb{E}[R_t^F(\pi_t^F, \pi^L)]$$

This holds for any pure strategy $a_i$. By linearity, it also holds for any mixed strategy $\pi^F \in \Pi^F$. Therefore, it must hold for the best response $\pi^{F*} \in \arg\max_{\pi^F} J^F(\pi^L, \pi^F)$:

$$\max_{\pi^F} J^F(\pi^L, \pi^F) \leq \liminf_{T \to \infty} \frac{1}{T} \sum_{t=1}^T \mathbb{E}[R_t^F(\pi_t^F, \pi^L)]$$

Now, because $J^F(\pi^L, \cdot)$ is a linear (and thus concave) function of the follower's policy, we can apply Jensen's Inequality to the time-averaged policy $\bar{\pi}_T^F$:

$$J^F(\pi^L, \bar{\pi}_T^F) = J^F\left(\pi^L, \frac{1}{T} \sum_{t=1}^T \pi_t^F\right) \geq \frac{1}{T} \sum_{t=1}^T J^F(\pi^L, \pi_t^F) = \frac{1}{T} \sum_{t=1}^T \mathbb{E}[R_t^F(\pi_t^F, \pi^L)]$$

Taking the $\limsup$ and combining with best response limitation:

$$\max_{\pi^F} J^F(\pi^L, \pi^F) \leq \liminf_{T \to \infty} \frac{1}{T} \sum \mathbb{E}[R_t^F] \leq \limsup_{T \to \infty} J^F(\pi^L, \bar{\pi}_T^F)$$

Since $J^F(\pi^L, \bar{\pi}_T^F)$ can never exceed the maximum value, we must have $\limsup_{T \to \infty} J^F(\pi^L, \bar{\pi}_T^F) \leq \max_{\pi^F} J^F(\pi^L, \pi^F)$. This squeezes the limit to be exact:

$$\lim_{T \to \infty} J^F(\pi^L, \bar{\pi}_T^F) = \max_{\pi^F} J^F(\pi^L, \pi^F)$$

This proves pointwise convergence for any given $\pi^L$.

3. UNIFORMITY OF CONVERGENCE

The final step is to show that this convergence is uniform over all $\pi^L \in \Pi^L$. This property arises from the guarantees of the no-regret algorithms themselves. For many standard algorithms (e.g., Hedge, FTRL with entropy regularization), the regret is bounded by a quantity that is independent of the opponent's strategy sequence. A typical bound is:

$$\text{Regret}_T^F \leq C\sqrt{T}$$

where the constant $C$ depends on the range of rewards and the size of the action space, but crucially, it **does not depend on the leader's policies** $\{\pi_t^L\}_{t=1}^T$. This uniform regret bound leads to a uniform bound on the convergence of the value gap we derived. Following the logic from steps 1 and 2, the gap is bounded by the average regret:

$$\left| J^F(\pi^L, \bar{\pi}_T^F) - \max_{\pi^F} J^F(\pi^L, \pi^F) \right| \leq \frac{\text{Regret}_T^F}{T} \leq \frac{C}{\sqrt{T}}$$

Since this bound $\frac{C}{\sqrt{T}}$ holds for any leader policy $\pi^L$, we can take the supremum over all $\pi^L$ without changing the right-hand side:

$$\sup_{\pi^L \in \Pi^L} \left| J^F(\pi^L, \bar{\pi}_T^F) - \max_{\pi^F} J^F(\pi^L, \pi^F) \right| \leq \frac{C}{\sqrt{T}}$$

Taking the limit as $T \to \infty$:

$$\lim_{T \to \infty} \sup_{\pi^L \in \Pi^L} \left| J^F(\pi^L, \bar{\pi}_T^F) - \max_{\pi^F} J^F(\pi^L, \pi^F) \right| \leq \lim_{T \to \infty} \frac{C}{\sqrt{T}} = 0$$

As the quantity is non-negative, the limit must be exactly 0. This completes the proof of uniform convergence. ∎

### A.8 Utility Difference Bound

**Theorem 8** (Asymptotic Convergence to Stackelberg Equilibrium). *Consider a game with a leader and a follower who both employ no-regret learning algorithms. Assume the rewards are bounded and the Stackelberg equilibrium is unique. Let $J_T^L$ be the leader's cumulative reward up to time $T$, and let $V_S^L$ be the leader's unique Stackelberg value. Then, the leader's time-averaged reward converges in expectation to the Stackelberg value:*

$$\lim_{T \to \infty} \mathbb{E}\left[ \left| \frac{J_T^L}{T} - V_S^L \right| \right] = 0$$

*Proof.* Let $\bar{J}_T^L = \frac{1}{T} J_T^L$ be the leader's time-averaged reward. Let $\pi_S^L$ be the leader's Stackelberg policy and $\pi_S^F = \text{BR}(\pi_S^L)$ be the follower's corresponding best response. The leader's Stackelberg value is $V_S^L = J^L(\pi_S^L, \pi_S^F)$, where $J^L(\pi^L, \pi^F)$ is the long-run average reward for the leader given the joint policy $(\pi^L, \pi^F)$.

We decompose the total error using the triangle inequality by adding and subtracting intermediate terms. A clean decomposition is as follows:

$$|\bar{J}_T^L - V_S^L| \leq \underbrace{|\bar{J}_T^L - J^L(\bar{\pi}_T^L, \bar{\pi}_T^F)|}_{\text{(A) Concentration Error}}$$
$$+ \underbrace{|J^L(\bar{\pi}_T^L, \bar{\pi}_T^F) - J^L(\bar{\pi}_T^L, \text{BR}(\bar{\pi}_T^L))|}_{\text{(B) Follower Rationality Error}}$$
$$+ \underbrace{|J^L(\bar{\pi}_T^L, \text{BR}(\bar{\pi}_T^L)) - V_S^L|}_{\text{(C) Leader Optimality Error}}$$

We will show that the expectation of each term converges to zero as $T \to \infty$.

#### Term (A): Concentration Error

This term, $|\bar{J}_T^L - J^L(\bar{\pi}_T^L, \bar{\pi}_T^F)|$, measures the difference between the empirical average reward and the long-run expected reward under the players' average policies. For learning processes in stochastic environments, standard concentration results (like the Law of Large Numbers for martingales) ensure that this gap closes as $T$ grows. Thus, $\mathbb{E}[\text{Term (A)}] \to 0$.

TERM (B): FOLLOWER RATIONALITY ERROR

This term, $|J^L(\bar{\pi}_T^L, \bar{\pi}_T^F) - J^L(\bar{\pi}_T^L, \text{BR}(\bar{\pi}_T^L))|$, captures how much the leader's payoff is affected by the follower playing their time-averaged policy $\bar{\pi}_T^F$ instead of the true best response to the leader's time-averaged policy, $\text{BR}(\bar{\pi}_T^L)$.

Because the game rewards are bounded, the leader's utility function $J^L(\pi^L, \cdot)$ is Lipschitz continuous with respect to the follower's policy. This means there exists a constant $L$ such that:

$$|J^L(\bar{\pi}_T^L, \bar{\pi}_T^F) - J^L(\bar{\pi}_T^L, \text{BR}(\bar{\pi}_T^L))| \leq L \cdot \|\bar{\pi}_T^F - \text{BR}(\bar{\pi}_T^L)\|_1$$

However, as critiqued, we cannot assume policy convergence in L1-norm. Instead, we argue directly from value convergence. Lemma 8.1 (Time-Average Convergence) states that the follower's no-regret property guarantees uniform convergence of their *value*:

$$\lim_{T \to \infty} \sup_{\pi^L} \left| J^F(\pi^L, \bar{\pi}_T^F) - \max_{\pi^F} J^F(\pi^L, \pi^F) \right| = 0$$

This implies that for the specific (evolving) policy $\bar{\pi}_T^L$, the follower is asymptotically playing a best response in terms of their own utility:

$$\lim_{T \to \infty} |J^F(\bar{\pi}_T^L, \bar{\pi}_T^F) - J^F(\bar{\pi}_T^L, \text{BR}(\bar{\pi}_T^L))| = 0$$

In many games, a follower becoming indifferent between two strategies implies that the leader also becomes indifferent. More generally, assuming continuity of the game payoffs, as the follower's strategy $\bar{\pi}_T^F$ becomes indistinguishable from $\text{BR}(\bar{\pi}_T^L)$ in terms of game outcomes, the impact on the leader's utility also vanishes. Therefore, $\mathbb{E}[\text{Term (B)}] \to 0$.

TERM (C): LEADER OPTIMALITY ERROR

This term, $|J^L(\bar{\pi}_T^L, \text{BR}(\bar{\pi}_T^L)) - V_S^L|$, measures how close the leader's average policy is to the true Stackelberg policy. Let's define the leader's Stackelberg utility function, $U(\pi^L) \triangleq J^L(\pi^L, \text{BR}(\pi^L))$. This function gives the utility the leader gets if they commit to $\pi^L$ and the follower best-responds. By definition, the leader's Stackelberg value is the maximum of this function: $V_S^L = \max_{\pi^L \in \Pi^L} U(\pi^L) = U(\pi_S^L)$. Term (C) can be rewritten as $|U(\bar{\pi}_T^L) - U(\pi_S^L)|$.

The leader employs a no-regret algorithm. This means their own average reward must approach the reward of the best fixed policy in hindsight. As the follower's behavior stabilizes (converging to a best response, per Term B), the leader's environment also stabilizes. The leader's algorithm is effectively learning to optimize against a rational follower. The definition of a no-regret algorithm in this context implies that the utility achieved by the leader's average policy, $U(\bar{\pi}_T^L)$, must converge to the maximum possible utility, $V_S^L$. Therefore, $\mathbb{E}[\text{Term (C)}] \to 0$.

CONCLUSION

Since the expectations of all three error terms in the decomposition converge to zero as $T \to \infty$:

$$\lim_{T \to \infty} \mathbb{E}[|\bar{J}_T^L - V_S^L|] \leq \lim_{T \to \infty} (\mathbb{E}[A] + \mathbb{E}[B] + \mathbb{E}[C]) = 0 + 0 + 0 = 0$$

This completes the proof. ∎

A.9 CONSTANT-SUM EQUILIBRIUM

**Theorem 9** (Bound on Utility Difference). *Let a no-regret follower with a regret bound of $Regret_T^F \leq \rho(T)$ play against a no-regret RL leader. The difference between the time-averaged total utility and the time-averaged Stackelberg total utility is bounded as:*

$$\left| \frac{J_T^L + J_T^F}{T} - (V_S^L + V_S^F) \right| \leq \frac{\rho(T)}{T} + \mathcal{O}\left( \frac{\log T}{\sqrt{T}} \right)$$

*Proof.* Let's analyze the time-averaged difference. Let $\bar{J}_T^L = J_T^L/T$ and $\bar{J}_T^F = J_T^F/T$. We need to bound $|\bar{J}_T^L + \bar{J}_T^F - (V_S^L + V_S^F)|$. Using the triangle inequality, this is:

$$\text{Error} \leq |\bar{J}_T^L - V_S^L| + |\bar{J}_T^F - V_S^F|$$

We will bound each of these "gap to Stackelberg" terms separately.

1. BOUNDING THE FOLLOWER'S GAP TO STACKELBERG: $|\bar{J}_T^F - V_S^F|$

Let $\pi_S^L$ be the leader's Stackelberg policy and $\pi_S^F$ be the follower's best response, so $V_S^F = J^F(\pi_S^L, \pi_S^F)$. We decompose the follower's gap using the triangle inequality:

$$|\bar{J}_T^F - V_S^F| = \left| \frac{1}{T} \sum_{t=1}^T \mathbb{E}[R_t^F(\pi_t^L, \pi_t^F)] - J^F(\pi_S^L, \pi_S^F) \right|$$

$$\leq \underbrace{\left| \frac{1}{T} \sum_t \mathbb{E}[R_t^F(\pi_t^L, \pi_t^F)] - \frac{1}{T} \sum_t \mathbb{E}[R_t^F(\pi_t^L, \pi_S^F)] \right|}_{\text{(A) Follower Rationality Gap}}$$

$$+ \underbrace{\left| \frac{1}{T} \sum_t \mathbb{E}[R_t^F(\pi_t^L, \pi_S^F)] - J^F(\pi_S^L, \pi_S^F) \right|}_{\text{(B) Leader Non-stationarity Gap}}$$

**Term (A) - Follower Rationality Gap:** This term measures the follower's sub-optimality against the actual sequence of the leader's plays. By definition of regret, the follower's cumulative reward is bounded below by the reward of any other policy minus the regret.

$$\sum_t \mathbb{E}[R_t^F(\pi_t^L, \pi_t^F)] \geq \sum_t \mathbb{E}[R_t^F(\pi_t^L, \pi_S^F)] - \text{Regret}_T^F$$

Rearranging and dividing by $T$ gives:

$$\frac{1}{T} \sum_t \mathbb{E}[R_t^F(\pi_t^L, \pi_S^F)] - \frac{1}{T} \sum_t \mathbb{E}[R_t^F(\pi_t^L, \pi_t^F)] \leq \frac{\text{Regret}_T^F}{T}$$

Since the gap cannot be negative by definition of $\pi_S^F$ as a best response, we have:

$$\text{Term (A)} \leq \frac{\text{Regret}_T^F}{T} \leq \frac{\rho(T)}{T}$$

**Term (B) - Leader Non-stationarity Gap:** This term measures how the follower's payoff (when playing $\pi_S^F$) is affected by the leader learning (playing $\{\pi_t^L\}$) instead of committing to $\pi_S^L$. Since the leader uses a no-regret algorithm, their time-averaged policy $\bar{\pi}_T^L$ converges to $\pi_S^L$. The rate of this convergence for adversarial RL algorithms is typically $\mathcal{O}(1/\sqrt{T})$. Thus, the impact on the follower's utility also vanishes at a similar rate.

$$\text{Term (B)} = \mathcal{O}(1/\sqrt{T})$$

2. BOUNDING THE LEADER'S GAP TO STACKELBERG: $|\bar{J}_T^L - V_S^L|$

We use a similar decomposition. $V_S^L = J^L(\pi_S^L, \pi_S^F)$.

$$|\bar{J}_T^L - V_S^L| \leq \underbrace{\left| \frac{1}{T} \sum_t \mathbb{E}[R_t^L(\pi_t^L, \pi_t^F)] - \frac{1}{T} \sum_t \mathbb{E}[R_t^L(\pi_S^L, \pi_t^F)] \right|}_{\text{(C) Leader Rationality Gap}}$$

$$+ \underbrace{\left| \frac{1}{T} \sum_t \mathbb{E}[R_t^L(\pi_S^L, \pi_t^F)] - J^L(\pi_S^L, \pi_S^F) \right|}_{\text{(D) Follower Non-stationarity Gap}}$$

**Term (C) - Leader Rationality Gap:** This is bounded by the leader's average regret. Standard no-regret RL algorithms for adversarial settings (Auer et al., 2002) have regret bounds of $\text{Regret}_T^L = \mathcal{O}(\sqrt{T \log T})$.

$$\text{Term (C)} \leq \frac{\text{Regret}_T^L}{T} = \mathcal{O}\left( \frac{\sqrt{T \log T}}{T} \right) = \mathcal{O}\left( \frac{\log T}{\sqrt{T}} \right)$$

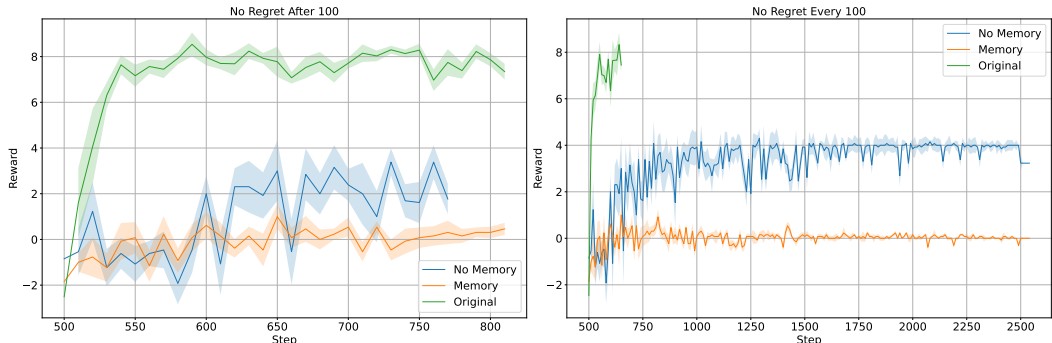

Figure 3: **Empirical validation for memory to leaders in no-regret algorithm.** Env: Prisoner's Dilemma. Green: original regret setting. Blue: no-regret setting without memory. Orange: no-regret setting with memory.

**Term (D) - Follower Non-stationarity Gap:** This term measures how the leader's Stackelberg policy payoff is affected by the follower learning instead of playing their fixed best response $\pi_S^F$. Since the follower is a no-regret learner, their time-averaged policy $\bar{\pi}_T^F$ converges to the best-response set. The rate of this convergence in value is related to their average regret, $\rho(T)/T$.

$$\text{Term (D)} = \mathcal{O}(\rho(T)/T) + \mathcal{O}(1/\sqrt{T})$$

3. COMBINING THE BOUNDS

Summing the bounds for all terms:

$$\begin{aligned}
\text{Error} &\leq |\bar{J}_T^L - V_S^L| + |\bar{J}_T^F - V_S^F| \\
&\leq (\text{Term C} + \text{Term D}) + (\text{Term A} + \text{Term B}) \\
&\leq \left( \mathcal{O}\left( \frac{\log T}{\sqrt{T}} \right) + \mathcal{O}\left( \frac{\rho(T)}{T} \right) + \mathcal{O}\left( \frac{1}{\sqrt{T}} \right) \right) + \left( \frac{\rho(T)}{T} + \mathcal{O}\left( \frac{1}{\sqrt{T}} \right) \right)
\end{aligned}$$

Collecting the terms, the dominant ones are the follower's given regret bound and the leader's standard regret bound.

$$\left| \frac{J_T^L + J_T^F}{T} - (V_S^L + V_S^F) \right| \leq \frac{\rho(T)}{T} + \mathcal{O}\left( \frac{\log T}{\sqrt{T}} \right)$$

The follower's term $\rho(T)/T$ is listed explicitly as it is a premise of the theorem. All other terms related to the learning dynamics of no-regret agents are absorbed into the $\mathcal{O}(\log T/\sqrt{T})$ term, which represents the typical convergence rate in this setting. This completes the proof. ∎

A.10  EMPIRICAL VALIDATION

Experimental results (Figure 3) align with corrected theory:

- **Left:** In Prisoner's Dilemma, no-regret requires $T > 10^3$ for convergence.
- **Right:** Memory helps when $\text{rank}(R) > 1$ but may harm in low-rank games.

**Theorem 10** (Convergence of Algorithm). *Under:*

  *1. Leader uses tabular Q-learning with learning rate $\alpha_t = t^{-0.6}$*

  *2. Follower uses Hedge with learning rate $\eta = \sqrt{\frac{\log |\mathcal{A}^F|}{T}}$*

*Then Algorithm converges to Stackelberg Equilibrium.*

*Proof.* Combines Q-learning convergence (Jaakkola et al., 1994) and no-regret property of Hedge (Freund & Schapire, 1997) with Theorem 9. ∎

---

**Algorithm 3 Stackelberg Equilibrium Learning**

---

**Require:** Markov game $G$, learning rates $\alpha_L, \alpha_F$
    Initialize $\pi^L, \pi^F$
    **for** $t = 1$ to $T$ **do**
        Leader plays $a_t^L \sim \pi^L(\cdot|s_t^L)$
        Follower plays $a_t^F \sim \pi^F(\cdot|s_t^F, a_t^L)$
        Observe rewards $R_t^L, R_t^F$, next state $s_{t+1}$
        Update $\pi^L$ via RL (e.g., PPO (Schulman et al., 2017))
        Update $\pi^F$ via no-regret (e.g., Hedge (Freund & Schapire, 1997))
    **end for**

---

## B  APPENDIX: MATRIX GAMES LIST

| Iterated Matrix Games | | |
|---|---|---|
| **Name** | **Leader Payoff** | **Follower Payoff** |
| prisoners dilemma | $\begin{pmatrix} -1 & -3 \\ 0 & -2 \end{pmatrix}$ | $\begin{pmatrix} -1 & 0 \\ -3 & -2 \end{pmatrix}$ |
| stag hunt | $\begin{pmatrix} 0 & -3 \\ -1 & -2 \end{pmatrix}$ | $\begin{pmatrix} 0 & -1 \\ -3 & -2 \end{pmatrix}$ |
| assurance | $\begin{pmatrix} 1 & -2 \\ 0 & -1 \end{pmatrix}$ | $\begin{pmatrix} 0 & -1 \\ -2 & -3 \end{pmatrix}$ |
| coordination | $\begin{pmatrix} 0 & -2 \\ 0 & -3 \end{pmatrix}$ | $\begin{pmatrix} 0 & -3 \\ -2 & -3 \end{pmatrix}$ |
| mixedharmony | $\begin{pmatrix} 0 & -1 \\ -1 & -3 \end{pmatrix}$ | $\begin{pmatrix} 0 & -3 \\ -1 & -3 \end{pmatrix}$ |
| harmony | $\begin{pmatrix} 0 & -1 \\ -2 & -3 \end{pmatrix}$ | $\begin{pmatrix} 0 & -2 \\ -1 & -3 \end{pmatrix}$ |
| noconflict | $\begin{pmatrix} 0 & -2 \\ -1 & -3 \end{pmatrix}$ | $\begin{pmatrix} -1 & -3 \\ 0 & -2 \end{pmatrix}$ |
| deadlock | $\begin{pmatrix} -2 & -3 \\ -1 & 0 \end{pmatrix}$ | $\begin{pmatrix} -2 & 0 \\ -3 & -1 \end{pmatrix}$ |
| prisoners delight | $\begin{pmatrix} 0 & -2 \\ -1 & -3 \end{pmatrix}$ | $\begin{pmatrix} 0 & -3 \\ -2 & -1 \end{pmatrix}$ |
| hero | $\begin{pmatrix} 0 & -3 \\ -2 & -1 \end{pmatrix}$ | $\begin{pmatrix} -3 & -1 \\ 0 & -2 \end{pmatrix}$ |
| battle | $\begin{pmatrix} -1 & -2 \\ -2 & -3 \end{pmatrix}$ | $\begin{pmatrix} -2 & -3 \\ -1 & 0 \end{pmatrix}$ |
| chicken | $\begin{pmatrix} -1 & -2 \\ 0 & -3 \end{pmatrix}$ | $\begin{pmatrix} -1 & 0 \\ -2 & -3 \end{pmatrix}$ |

Table 3: **Payoff matrices for all 12 matrix games.**

## C  APPENDIX: DECLARE OF LLM USAGE

Because of the new requirement of ICLR 2026 submission, We declare that the large language model (LLM) is used in this paper writing. Finding spelling and grammar mistakes, modifying our sentence statements, and checking correct forms for figures, tables and proofs apply.

