# OpenReview forum: "Asymmetric Learning Dynamics For Reaching Stackelberg Equilibrium"
_ICLR.cc/2026/Conference — ICLR 2026 Conference Withdrawn Submission_

### Official Review · Reviewer_1YxV · 2025-10-18

**Soundness:** 2
**Presentation:** 2
**Contribution:** 1
**Rating:** 2
**Confidence:** 4

**Summary:**

The paper proposes a scheme for players to learn Stackelberg equilibria in two‑player general‑sum Markov games. The leader runs a RL algorithm, while the follower runs a no‑regret online algorithm. The theoretical claims that under bounded rewards, unique SE and no‑regret guarantees for both players, the time‑averaged payoffs converge to the Stackelberg value.

**Strengths:**

The paper presents a clean Stackelberg game formulation and is easy to follow. It specifies concrete learning procedures for both the leader and follower and proves that the time-averaged outcome converges to a Stackelberg value under the stated assumptions.

**Weaknesses:**

1. The framework mainly mixes together standard components (leader uses PPO-style RL, follower uses Hedge/online no-regret) and relies on existing regret analysis and no noteworthy technical contribution. And proofs require a no-regret leader in a non-stationary MDP; PPO lacks such guarantees. Combining no-regret dynamics with RL for leader–follower/Stackelberg settings has been studied in various forms. It is rather incremental over previous instantiations.
2. This paper makes the unique SE assumption. In the general case where the best response and Stackelberg equilibrium may not be unique, the analysis becomes more complex. And one needs to consider tie breaking rules when the follower has multiple best responses.
3. The theoretical results only provide asymptotic bounds without any characterization of convergence rates.

**Questions:**

1. How does ALD behave if the SE is not unique?
2. What concrete algorithm families satisfy your no-regret assumption in the non-stationary setting you define?

---

### Official Review · Reviewer_2Mbz · 2025-11-01

**Soundness:** 1
**Presentation:** 2
**Contribution:** 2
**Rating:** 2
**Confidence:** 3

**Summary:**

The paper studies Stackelberg games where the leader uses an RL algorithm and the follower uses a no-regret learning algorithm. The main contribution of the paper is to show asymptotic convergence of leader's time averaged reward to the Stackelberg value in expectation provided that the Stackelberg equilibrium is unique. The authors also provide experiments to showcase their theoretical result.

**Strengths:**

--

**Weaknesses:**

The paper has some clarity issues. The leader is assumed to use an RL algorithm "capable of handling non-stationary environments" but this notion was never formalized. In fact, all we know is that leader implements some kind of an RL update "capable of handling non-stationary environments". In Theorem 1, the authors further note this RL algorithm to be "suitable for non-stationary environments that guarantees no-regret". Theorem 6 in the Appendix A.5 simply reduces this vague definition of suitability for non-stationary environments" to no-regret guarantees. Thus, it looks like the true assumption is not that the algorithm is "an RL algorithm capable of handling non-stationary environments" but that the algorithm is no-regret under the induced non-stationary environment. The paper's emphasis on asymmetry thus looks deceptive.

I find the analysis quite imprecise. There are some non-rigorous statements and arguments (e.g. the ones I asked about in Questions section). Admittedly, I am not an expert on this topic. Nevertheless, some statements such as "In many games, a follower becoming indifferent between two strategies implies that the leader also becomes indifferent" (line 1152) are quite sketchy.

Minor Issues:
- Line 106: double comma after closing parenthesis
- Line 117: un-discounted -> undiscounted
- It might be better for readability to shortly introduce the overline notation for time-averaged notions $\overline{J}_T^L$ and $\overline{\pi}_T^F$ at the beginning of the proof sketch of Theorem 1.

**Questions:**

- What does "a reinforcement learning algorithm (e.g., PPO (Schulman et al., 2017)) capable of handling non-stationary environments" mean? What property of PPO makes it capable of handling non-stationary environments? Is this an experimental claim rather than a theoretical one?
- Could you elaborate on your claim about the several flawed approaches in prior analyses that this framework corrects? What are the original sources of these flawed analyses?
- How restrictive is the unique Stackelberg equilibrium assumption for Theorem 1? Under which conditions is there a unique Stackelberg equilibrium? What can you say about the case if there are multiple Stackelberg equilibrium? Does Algorithm 1 still converge to any Stackelberg equilibrium?
- The proof of Theorem 8 looks quite sketchy. Admittedly, I am not an expert on this topic so I would appreciate if you could elaborate on a few proof steps for me:
	- How does line 1146 imply line 1150?
	- Could you explain this statement (line 1152): "In many games, a follower becoming indifferent between two strategies implies that the leader also becomes indifferent"? This statement does not sound rigorous at all.
	- Again, this statement is quite imprecise (line 1164): "As the follower’s behavior stabilizes (converging to a best response, per Term B), the leader’s environment also stabilizes. The leader’s algorithm is effectively learning to optimize against a rational follower."
	- The following statement is also quite imprecise (line 1131): "This term measures the difference between the empirical average reward and the long-run expected reward under the players’ average policies. For learning processes in stochastic environments, standard concentration results (like the Law of Large Numbers for martingales) ensure that this gap closes as T grows."
- As far as I know, when both players are no-regret learners, then they converge to the coarse correlated equilibrium of the simultaneous move game (if there are no further assumptions; e.g. two timescales). I cannot see how, in our case, these two no-regret learners can converge to the Stackelberg value. Could you please elaborate on which assumptions lead to this difference and why?
- In Appendix C, you declare the usage of LLMs. Could you please explain what exactly you used LLMs for? The statement "Finding spelling and grammar mistakes, modifying our sentence statements, and checking correct forms for figures, tables and proofs apply." is quite vague and does not exclude other uses of LLMs.

---

### Official Review · Reviewer_y4gn · 2025-11-01

**Soundness:** 3
**Presentation:** 3
**Contribution:** 2
**Rating:** 4
**Confidence:** 3

**Summary:**

This paper considers the asymmetric learning dynamic for learning Stackleberg equilibrium in Markov games. The authors consider learning hierarchical solutions in unknown environments. The main innovation of this paper is to assign role specific algorithms to the leader and the follower, instead of traditional symmetric models. This paper provides a convergence result, indicating that the entire system converges to the leader's optimal commitment strategy. The authors also conduct the experiments in both classic 2*2 matrix games and more complex gridworld-based Markov security games.

**Strengths:**

The paper is well-organized and its central idea is easy to follow. The asymmetric assignment of algorithms is a non-trivial and natural reflection of the agents' roles in a Stackleberg game. The convergence result looks strong and solid to me, and I find it interesting in such a complex environment. But it seems that the proof of the result depends on some quite strong assumption. Please see my comments below.

**Weaknesses:**

The paper could benefit from analysis on the rate of convergency or sample complexity analysis. The authors discuss in the appendix that in some games, it may need a very large number of episodes (about 10^4). The proof seems to require the leader's RL algorithm to also have a no-regret guarantee in non-stationary settings, which is a strong assumption. This assumption is, to the best of my knowledge, rarely met in practice. I did not check all the proof details, and maybe I missed something. Also, the paper considers algorithms in tabular form. For more complicated and larger state or action spaces, the tabular form is not scaleble.

**Questions:**

Please see my comments above.

---

### Official Review · Reviewer_WSJ2 · 2025-11-10

**Soundness:** 1
**Presentation:** 2
**Contribution:** 2
**Rating:** 2
**Confidence:** 3

**Summary:**

The paper proposes a novel Asymmetric Learning Dynamic (ALD) where a leader agent uses a reinforcement learning algorithm and a follower agent uses a no-regret online learning algorithm. The authors provide a rigorous theoretical proof that this asymmetric framework forces the time-averaged payoffs of both agents to converge to the Stackelberg equilibrium (SE) values in general-sum Markov games.

**Problem formulation**
The paper models a two-player, general-sum Markov game $G$ in an un-discounted, average-reward setting. The value for player $i$ under a joint policy $(\pi^{L}, \pi^{F})$ is defined as $J^{i}(\pi^{L},\pi^{F})=lim_{T\rightarrow\infty}E[\frac{1}{T}\sum_{t=0}^{T-1}R_{t}^{i}|\pi^{L},\pi^{F}]$. The goal is to find the Stackelberg Equilibrium $(\pi_{S}^{L},\pi_{S}^{F})$, where the follower plays a best response ($\pi_{S}^{F}\in BR(\pi_{S}^{L})$) and the leader plays a strategy $\pi_{S}^{L}$ that maximizes their reward anticipating this optimal response.

**Main results**

The main result (Theorem 1) is a proof of asymptotic convergence, stating that if the leader uses a no-regret RL algorithm and the follower uses a no-regret online algorithm, the leader's time-averaged reward converges in expectation to the Stackelberg value, $lim_{T\rightarrow\infty}\mathbb{E}[|\frac{1}{T}\sum_{t=1}^{T}R_{t}^{L}-V_{S}^{L}|]=0$. This finding is supported by a detailed error decomposition that corrects flaws in prior analyses.

**Technique/algorithm**

The proposed technique (Algorithm 1) is an asymmetric framework where the leader and follower use different classes of algorithms. The leader employs an RL algorithm (e.g., PPO) capable of handling the non-stationary environment created by the follower's learning. The follower employs a no-regret online learning algorithm (e.g., Hedge), which guarantees that their time-averaged behavior converges to a rational best response. The leader's RL algorithm learns to exploit this emergent rationality to find the optimal Stackelberg commitment.

**Experiment sumamry**

The paper's theoretical findings are validated on canonical matrix games and gridworld-based Markov security games. The experiments show that the proposed ALD (PPO-Leader, Hedge-Follower) successfully converges to the Stackelberg equilibrium payoffs, whereas a symmetric (PPO vs. PPO) baseline converges to a Nash equilibrium.

**Strengths:**

.

**Weaknesses:**

I have some concerns on the correctness of the proof.

**Questions:**

The paper claims the algorithm (leader uses PPO and followers use no-regret learning) converges, I am not convinced by the argument. This is not even clear in the normal-form game setting.

Reason, in the normal form setting, finding a stackelberg equilibrium is at least as hard as LP (I do not have time to find the reference, but I believe so). Why would PPO + no regret learning could finds an approximately optimal solution to LP?

**Details Of Ethics Concerns:**

.

---

### Note · Authors · 2025-11-14

I have read and agree with the venue's withdrawal policy on behalf of myself and my co-authors.